# ON DATA-AUGMENTATION AND CONSISTENCY-BASED SEMI-SUPERVISED LEARNING

**Atin Ghosh & Alexandre H. Thiery**
Department of Statistics and Applied Probability
National University of Singapore
`atin.ghosh@u.nus.edu`
`a.h.thiery@nus.edu.sg`

## ABSTRACT

Recently proposed consistency-based Semi-Supervised Learning (SSL) methods such as the $\Pi$-model, temporal ensembling, the mean teacher, or the virtual adversarial training, have advanced the state of the art in several SSL tasks. These methods can typically reach performances that are comparable to their fully supervised counterparts while using only a fraction of labelled examples. Despite these methodological advances, the understanding of these methods is still relatively limited. In this text, we analyse (variations of) the $\Pi$-model in settings where analytically tractable results can be obtained. We establish links with Manifold Tangent Classifiers and demonstrate that the quality of the perturbations is key to obtaining reasonable SSL performances. Importantly, we propose a simple extension of the Hidden Manifold Model that naturally incorporates data-augmentation schemes and offers a framework for understanding and experimenting with SSL methods.

## 1 INTRODUCTION

Consider a dataset $\mathcal{D} = \mathcal{D}_L \cup \mathcal{D}_U$ that is comprised of labelled samples $\mathcal{D}_L = \{x_i, y_i\}_{i \in \mathbf{I}_L}$ as well as unlabelled samples $\mathcal{D}_U = \{x_i\}_{i \in \mathbf{I}_U}$. Semi-Supervised Learning (SSL) is concerned with the use of both the labelled and unlabeled data for training. In many scenarios, collecting labelled data is difficult or time consuming or expensive so that the amount of labelled data can be relatively small when compared to the amount of unlabelled data. The main challenge of SSL is in the design of methods that can exploit the information contained in the distribution of the unlabelled data (Zhu05; CSZ09).

In modern high-dimensional settings that are common to computer vision, signal processing, Natural Language Processing (NLP) or genomics, standard graph/distance based methods (BC01; ZG02; ZGL03; BNS06; DSST19) that are successful in low-dimensional scenarios are difficult to implement. Indeed, in high-dimensional spaces, it is often difficult to design sensible notions of distances that can be exploited within these methods. We refer the interested reader to the book-length treatments (Zhu05; CSZ09) for discussion of other approaches.

The *manifold assumption* is the fundamental structural property that is exploited in most modern approaches to SSL: high-dimensional data samples lie in a small neighbourhood of a low-dimensional manifold (TP91; BJ03; Pey09; Cay05; RDV$^+$11). In computer vision, the presence of this low-dimensional structure is instrumental to the success of (variational) autoencoder and generative adversarial networks: large datasets of images can often be parametrized by a relatively small number of degrees of freedom. Exploiting the unlabelled data to uncover this low-dimensional structure is crucial to the design of efficient SSL methods. A recent and independent evaluation of several modern methods for SSL can be found in (OOR$^+$18). It is found there that consistency-based methods (BAP14; SJT16; LA16; TV17; MMIK18; LZL$^+$18; GSA$^+$20), the topic of this paper, achieve state-of-the art performances in many realistic scenarios.

**Contributions:** consistency-based semi-supervised learning methods have recently been shown to achieve state-of-the-art results. Despite these methodological advances, the understanding of these methods is still relatively limited when compared to the fully-supervised setting (SMG13;

AS17; SBD$^+$18; TZ15; SZT17). In this article, we do not propose a new SSL method. Instead, we analyse consistency-based methods in settings where analytically tractable results can be obtained, when the data-samples lie in the neighbourhood of well-defined and tractable low-dimensional manifolds, and simple and controlled experiments can be carried out. We establish links with Manifold Tangent Classifiers and demonstrate that consistency-based SSL methods are in general more powerful since they can better exploit the local geometry of the data-manifold if efficient data-augmentation/perturbation schemes are used. Furthermore, in section 4.1 we show that the popular *Mean Teacher* method and the conceptually more simple $\Pi$-model approach share the same solutions in the regime when the data-augmentations are small; this confirms often reported claim that the data-augmentation schemes leveraged by the recent SSL, as well as fully unsupervised algorithms, are instrumental to their success. Finally, in section 4.3 we propose an extension of the Hidden Manifold Model (GMKZ19; GLK$^+$20). This generative model allows us to investigate the properties of consistency-based SSL methods, taking into account the data-augmentation process and the underlying low-dimensionality of the data, in a simple and principled manner, and without relying on a specific dataset. For gaining understanding of SSL, as well as self-supervised learning methods, we believe it to be important to develop a framework that **(i)** can take into account the geometry of the data **(ii)** allows the study of the influence of the quality of the data-augmentation schemes **(iii)** does not rely on any particular dataset. While the understanding of fully-supervised methods have largely been driven by the analysis of simplified model architectures (eg. linear and two-layered models, large dimension asymptotic such as the Neural Tangent Kernel), these analytical tools alone are unlikely to be enough to explain the mechanisms responsible for the success of SSL and self-supervised learning methods (CKNH20; GSA$^+$20), since they do not, and cannot easily be extended to, account for the geometry of the data and data-augmentation schemes. Our proposed framework offers a small step in that direction.

## 2 CONSISTENCY-BASED SEMI-SUPERVISED LEARNING

For concreteness and clarity of exposition, we focus the discussion on classification problems. The arguments described in the remaining of this article can be adapted without any difficulty to other situations such as regression or image segmentation. Assume that the samples $x_i \in \mathcal{X} \subset \mathbb{R}^D$ can be represented as $D$-dimensional vectors and that the labels belong to $C \geq 2$ possible classes, $y_i \in \mathcal{Y} \equiv \{1, \ldots, C\}$. Consider a mapping $\mathcal{F}_\theta : \mathbb{R}^D \to \mathbb{R}^C$ parametrized by $\theta \in \Theta \subset \mathbb{R}^{|\Theta|}$. This can be a neural network, although that is not necessary. For $x \in \mathcal{X}$, the quantity $\mathcal{F}_\theta(x)$ can represent probabilistic output of the classifier, or , for example, the pre-softmax activations. Empirical risk minimization consists in minimizing the function

$$\mathcal{L}_L(\theta) = \frac{1}{|\mathcal{D}_L|} \sum_{i \in \mathbf{I}_L} \ell(\mathcal{F}_\theta(x_i), y_i)$$

for a loss function $\ell : \mathbb{R}^C \times \mathcal{Y} \mapsto \mathbb{R}$. Maximum likelihood estimation corresponds to choosing the loss function as the cross entropy. The optimal parameter $\theta \in \Theta$ is found by a variant of stochastic gradient descent (RM51) with estimated gradient

$$\nabla_\theta \left\{ \frac{1}{|\mathcal{B}_L|} \sum_{i \in \mathcal{B}_L} \ell(\mathcal{F}_\theta(x_i), y_i) \right\}$$

for a mini-batch $\mathcal{B}_L$ of labelled samples. Consistency-based SSL algorithms regularize the learning by enforcing that the learned function $x \mapsto \mathcal{F}_\theta(x)$ respects local derivative and invariance constraints. For simplicity, assume that the mapping $x \mapsto \mathcal{F}_\theta(x)$ is deterministic, although the use of dropout (SHK$^+$14) and other sources of stochasticity are popular in practice. The $\Pi$-model (LA16; SJT16) makes use of a stochastic mapping $\mathcal{S} : \mathcal{X} \times \Omega \to \mathcal{X}$ that maps a sample $x \in \mathcal{X}$ and a source of randomness $\omega \in \Omega \subset \mathbb{R}^{d_\Omega}$ to another sample $\mathcal{S}_\omega(x) \in \mathcal{X}$. The mapping $\mathcal{S}$ describes a stochastic *data augmentation* process. In computer vision, popular data-augmentation schemes include random translations, rotations, dilatations, croppings, flippings, elastic deformations, color jittering, addition of speckle noise, and many more domain-specific variants. In NLP, synonym replacements, insertions and deletions, back-translations are often used although it is often more difficult to implement these data-augmentation strategies. In a purely supervised setting, data-augmentation can be used as a

regularizer. Instead of directly minimizing $\mathcal{L}_L$, one can minimize instead

$$\theta \mapsto \frac{1}{|\mathcal{D}_L|} \sum_{i \in \mathbf{I}_L} \mathbb{E}_\omega[\ell(\mathcal{F}_\theta[\mathbb{S}_\omega(x_i)], y_i)].$$

In practice, data-augmentation regularization, although a simple strategy, is often crucial to obtaining good generalization properties (PW17; CZM$^+$18; LBC17; PCZ$^+$19). The idea of regularizing by enforcing robustness to the injection of noise can be traced back at least to (Bis95). In the $\Pi$-model, the data-augmentation mapping $\mathbb{S}$ is used to define a consistency regularization term,

$$\mathcal{R}(\theta) = \frac{1}{|\mathcal{D}|} \sum_{i \in \mathbf{I}_L \cup \mathbf{I}_U} \mathbb{E}_\omega \left\{ \left\| \mathcal{F}_\theta[\mathbb{S}_\omega(x_i)] - \mathcal{F}_{\theta_\star}(x_i) \right\|^2 \right\}. \tag{1}$$

The notation $\theta_\star$ designates a copy of the parameter $\theta$, i.e. $\theta_\star = \theta$, and emphasizes that when differentiating the consistency regularization term $\theta \mapsto \mathcal{R}(\theta)$, one does not differentiate through $\theta_\star$. In practice, a stochastic estimate of $\nabla \mathcal{R}(\theta)$ is obtained as follows. For a mini-batch $\mathcal{B}$ of samples $\{x_i\}_{i \in \mathcal{B}}$, the current value $\theta_\star \in \Theta$ of the parameter and the current predictions $f_i \equiv \mathcal{F}_{\theta_\star}(x_i)$, the quantity

$$\nabla \left\{ \frac{1}{|\mathcal{B}|} \sum_{i \in \mathcal{B}} \left\| \mathcal{F}_\theta[\mathbb{S}_\omega(x_i)] - f_i \right\|^2 \right\}$$

is an approximation of $\nabla \mathcal{R}(\theta)$. There are indeed many variants (eg. use of different norms, different manners to inject noise), but the general idea is to force the learned function $x \mapsto \mathcal{F}_\theta(x)$ to be locally invariant to the data-augmentation scheme $\mathbb{S}$. Several extensions such as the Mean Teacher (TV17) and the VAT (MMIK18) schemes have been recently proposed and have been shown to lead to good results in many SSL tasks. The recently proposed and state-of-the-art BYOL approach (GSA$^+$20) is relying on mechanisms that are very close to the consistency regularization methods discussed on this text.

If one recalls the manifold assumption, this approach is natural: since the samples corresponding to different classes lie on separate manifolds, the function $\mathcal{F}_\theta : \mathcal{X} \to \mathbb{R}^C$ should be constant on each one of these manifolds. Since the correct value of $\mathcal{F}_\theta$ is typically well approximated or known for labelled samples $(x_i, y_i) \in \mathcal{D}_L$, the consistency regularization term equation 1 helps propagating these known values across these manifolds. This mechanism is indeed similar to standard SSL graph-based approaches such as label propagation (ZG02). Graph-based methods are difficult to directly implement in computer vision, or NLP, when a meaningful notion of distance is not available. This interpretation reveals that it is crucial to include the labelled samples in the regularization term equation 1 in order to help *propagating* the information contained in the labelled samples to the unlabelled samples. Our numerical experiments suggest that, in the standard setting when the number of labelled samples is much lower than the number of unlabeled samples, i.e. $|\mathcal{D}_L| \ll |\mathcal{D}_U|$, the formulation equation 1 of the consistency regularization leads to sub-optimal results and convergence issues: the information contained in the labelled data is swamped by the number of unlabelled samples. In all our experiments, we have adopted instead the following regularization term

$$\begin{aligned} \mathcal{R}(\theta) = {} & \frac{1}{|\mathcal{D}_L|} \sum_{i \in \mathbf{I}_L} \mathbb{E}_\omega \left\{ \left\| \mathcal{F}_\theta[\mathbb{S}_\omega(x_i)] - \mathcal{F}_{\theta_\star}(x_i) \right\|^2 \right\} \\ & + \frac{1}{|\mathcal{D}_U|} \sum_{j \in \mathbf{I}_U} \mathbb{E}_\omega \left\{ \left\| \mathcal{F}_\theta[\mathbb{S}_\omega(x_j)] - \mathcal{F}_{\theta_\star}(x_j) \right\|^2 \right\} \end{aligned} \tag{2}$$

that balances the labelled and unlabelled data samples more efficiently. Furthermore, it is clear that the quality and variety of the data-augmentation scheme $\mathbb{S} : \mathcal{X} \times \Omega \to \mathcal{X}$ is pivotal to the success of consistency-based SSL methods. We argue in this article that it is the dominant factor contributing to the success of this class of methods. Effort spent on building efficient local data-augmentation schemes will be rewarded in terms of generalization performances. Designing good data-augmentation schemes is an efficient manner of injecting expert/prior knowledge into the learning process. It is done by leveraging the understanding of the local geometry of the data manifold. As usual and not surprisingly (NGP98; MHF$^+$12), in data-scarce settings, any type of domain-knowledge needs to be exploited and we argue that consistency regularization approaches to SSL are instances of this general principle.

## 3   Approximate Manifold Tangent Classifier

It has long been known (SLDV98) that exploiting the knowledge of derivatives, or more generally enforcing local invariance properties, can greatly enhance the performance of standard classifiers/regressors (HK02; CS02). In the context of deep-learning, the Manifold Tangent Classifier (RDV$^+$11) is yet another illustration of this idea. Consider the data manifold $\mathcal{M} \subset \mathcal{X} \subset \mathbb{R}^D$ and assume that the data samples lie on a neighbourhood of it. For $x \in \mathcal{M}$, consider as well the tangent plane $T_x$ to $\mathcal{M}$ at $x$. Assuming that the manifold $\mathcal{M}$ is of dimension $1 \leq d \leq D$, the tangent plane $T_x$ is also of dimension $d$ with an orthonormal basis $\mathbf{e}_1^x, \ldots, \mathbf{e}_d^x \in \mathbb{R}^D$. This informally means that, for suitably small coefficients $\omega_1, \ldots, \omega_d \in \mathbb{R}$, the transformed sample $\overline{x} \in \mathcal{X}$ defined as

$$\overline{x} \;=\; x + \sum_{j=1}^{d} \omega_j \, \mathbf{e}_j^x$$

also lies, or is very close to, the data manifold $\mathcal{M}$. A possible stochastic data-augmentation scheme can therefore be defined as $\mathcal{S}_\omega(x) = x + V_\omega$ where $V_\omega = \sum_{j=1}^{d} \omega_j \, \mathbf{e}_j^x$. If $\omega$ is a multivariate $d$-dimensional centred Gaussian random vector with suitably small covariance matrix, the perturbation vector $V_\omega$ is also centred and normally distributed. To enforce that the function $x \to \mathcal{F}_\theta(x)$ is locally approximately constant along the manifold $\mathcal{M}$, one can thus penalize the derivatives of $\mathcal{F}_\theta$ at $x$ in the directions $V_\omega$. Denoting by $\mathbf{J}_x \in \mathbb{R}^{C,D}$ the Jacobian with respect to $x \in \mathbb{R}^D$ of $\mathcal{F}_\theta$ at $x \in \mathcal{M}$, this can be implemented by adding a penalization term of the type $\mathbb{E}_\omega[\|\mathbf{J}_x \, V_\omega\|^2] = \mathbf{Tr}\big(\Gamma \otimes \mathbf{J}_x^T \mathbf{J}_x\big)$, where $\Gamma \in \mathbb{R}^{D,D}$ is the covariance matrix of the random vector $\omega \to V_\omega$. This type of regularization of the Jacobian along the data-manifold is for example used in (BNS06). More generally, if one assumes that for any $x, \omega \in \mathcal{X} \times \Omega$ we have $\mathcal{S}_{\varepsilon \, \omega}(x) = x + \varepsilon \, \mathbf{D}(x, \omega) + \mathcal{O}(\varepsilon^2)$, for some derivative mapping $\mathbf{D} : \mathcal{X} \times \Omega \to \mathcal{X}$, it follows that

$$\lim_{\varepsilon \to 0} \frac{1}{\varepsilon^2} \, \mathbb{E}_\omega\big[\|\mathcal{F}_\theta[\mathcal{S}_{\varepsilon \, \omega}(x)] - \mathcal{F}_\theta(x)\|^2\big] \;=\; \mathbb{E}_\omega\big[\|\mathbf{J}_x \, \mathbf{D}(x, \omega)\|^2\big] \;=\; \mathbf{Tr}\big(\Gamma_{x, \mathcal{S}} \otimes \mathbf{J}_x^T \, \mathbf{J}_x\big)$$

where $\Gamma_{x, \mathcal{S}}$ is the covariance matrix of the $\mathcal{X}$-valued random vector $\omega \mapsto \mathbf{D}(x, \omega) \in \mathcal{X}$. This shows that consistency-based methods can be understood as approximated Jacobian regularization methods, as proposed in (SLDV98; RDV$^+$11).

### 3.1   Limitations

In practice, even if many local dimension reduction techniques have been proposed, it is still relatively difficult to obtain a good parametrization of the data manifold. The Manifold Tangent Classifier (MTC) (RDV$^+$11) implements this idea by first extracting in an unsupervised manner a good representation of the dataset $\mathcal{D}$ by using a Contractive-Auto-Encoder (CAE) (RVM$^+$11). This CAE can subsequently be leveraged to obtain an approximate basis of each tangent plane $T_{x_i}$ for $x_i \in \mathcal{D}$, which can then be used for penalizing the Jacobian of the mapping $x \mapsto \mathcal{F}_\theta(x)$ in the direction of the tangent plane to $\mathcal{M}$ at $x$. The above discussion shows that the somewhat simplistic approach consisting in adding an isotropic Gaussian noise to the data samples is unlikely to deliver satisfying results. It is equivalent to penalizing the Frobenius norm $\|\mathbf{J}_x\|_{\mathrm{F}}^2$ of the Jacobian of the mapping $x \mapsto \mathcal{F}_\theta(x)$; in a linear model, that is equivalent to the standard *ridge regularization*. This mechanism does not take at all into account the local-geometry of the data-manifold. Nevertheless, in medical imaging applications where scans are often contaminated by speckle noise, this class of approaches which can be thought off as adding artificial speckle noise, can help mitigate over-fitting (DRS$^+$18).

There are many situations where, because of data scarcity or the sheer difficulty of unsupervised representation learning in general, domain-specific data-augmentation schemes lead to much better regularization than Jacobian penalization. Furthermore, as schematically illustrated in Figure 1, Jacobian penalization techniques are not efficient at learning highly non-linear manifolds that are common, for example, in computer vision. For example, in "pixel space", a simple image translation is a highly non-linear transformation only well approximated by a first order approximation for very small translations. In other words, if $x \in \mathcal{X}$ represents an image and $g(x, v)$ is its translated version by a vector $v$, the approximation $g(x, v) \approx x + \nabla_v g(x)$, with $\nabla_v g(x) \equiv \lim_{\varepsilon \to 0} (g(x, \varepsilon \, v) - g(x)/\varepsilon$, becomes poor as soon as the translation vector $v$ is not extremely small.

In computer vision, translations, rotations and dilatations are often used as sole data-augmentation schemes: this leads to a poor local exploration of the data-manifold since this type transformations

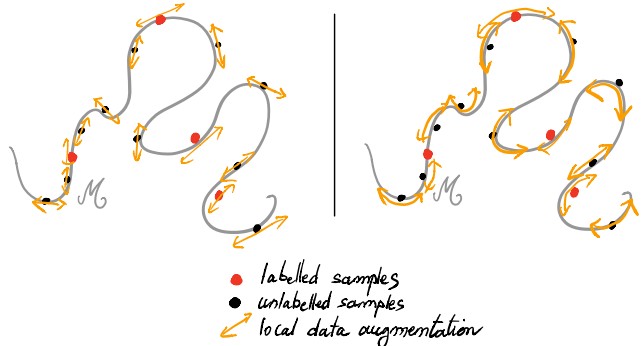

Figure 1: **Left:** Jacobian (i.e. first order) Penalization method are short-sighted and do not exploit fully the data-manifold **Right:** Data-Augmentation respecting the geometry of the data-manifold.

only generate a very low dimensional exploration manifold. More precisely, the exploration manifold emanating from a sample $x_0 \in \mathcal{X}$, i.e. $\{\mathcal{S}(x_0, \omega) \; : \; \omega \in \Omega\}$, is very low dimensional: its dimension is much lower than the dimension $d$ of the data-manifold $\mathcal{M}$. Enriching the set of data-augmentation degrees of freedom with transformations such as elastic deformation or non-linear pixel intensity shifts is crucial to obtaining a high-dimensional local exploration manifold that can help propagating the information on the data-manifold efficiently (CZM$^+$19; PCZ$^+$19).

## 4 ASYMPTOTIC PROPERTIES

### 4.1 FLUID LIMIT

Consider the standard $\Pi$-model trained with a standard Stochastic Gradient Descent (SGD). Denote by $\theta_t \in \Theta$ the current value of the parameter and $\eta > 0$ the learning rate. We have

$$\theta_{k+1} \; = \; \theta_k - \eta \, \nabla_\theta \bigg\{ \frac{1}{|\mathcal{B}_L|} \sum_{i \in \mathcal{B}_L} \ell(\, \mathcal{F}_{\theta_k}(x_i), \, y_i \,) + \frac{\lambda}{|\mathcal{B}_L|} \sum_{j \in \mathcal{B}_L} \left\| \mathcal{F}_{\theta_k}(\mathcal{S}_\omega[x_j]) - f_j \right\|^2$$
$$+ \frac{\lambda}{|\mathcal{B}_U|} \sum_{k \in \mathcal{B}_U} \left\| \mathcal{F}_{\theta_k}(\mathcal{S}_\omega[x_k]) - f_k \right\|^2 \bigg\} \tag{3}$$

for a parameter $\lambda > 0$ that controls the trade-off between supervised and consistency losses, as well as subsets $\mathcal{B}_L$ and $\mathcal{B}_U$ of labelled and unlabelled data samples, and $f_j \equiv \mathcal{F}_{\theta_\star}(x_j)$ for $\theta_\star \equiv \theta_k$ as discussed in Section 2. The right-hand-side is an unbiased estimate of $\eta \, \nabla_\theta \big[ \mathcal{L}_L(\theta_k) + \lambda \, \mathcal{R}(\theta_k) \big]$ with variance of order $\mathcal{O}(\eta^2)$, where the regularization term $\mathcal{R}(\theta_k)$ is described in equation 2. It follows from standard fluid limit approximations (EK09)[Section 4.8] for Markov processes that, under mild regularity and growth assumptions and as $\eta \to 0$, the appropriately time-rescaled trajectory $\{\theta_k\}_{k \geq 0}$ can be approximated by the trajectory of the Ordinary Differential Equation (ODE).

**Proposition 4.1** *Let* $\mathbf{D}([0, T], \mathbb{R}^{|\Theta|})$ *be the usual space of càdlàg* $\mathbb{R}^{|\Theta|}$*-valued functions on a bounded time interval* $[0, T]$ *endowed with the standard Skorohod topology. Consider the update equation 3 with learning rate* $\eta > 0$ *and define the continuous time process* $\overline{\theta}^\eta(t) = \theta_{[t/\eta]}$. *The sequence of processes* $\overline{\theta}^\eta \in \mathbf{D}([0, T], \mathbb{R}^{|\Theta|})$ *converges weakly in* $\mathbf{D}([0, T], \mathbb{R}^{|\Theta|})$ *and as* $\eta \to 0$ *to the solution of the ordinary differential equation*

$$\dot{\overline{\theta}}_t \; = \; -\nabla \Big( \mathcal{L}(\overline{\theta}_t) + \lambda \, \mathcal{R}(\overline{\theta}_t) \Big). \tag{4}$$

The article (TV17) proposes the *mean teacher* model, an averaging approach related to the standard Polyak-Ruppert averaging scheme (Pol90; PJ92), which modifies the consistency regularization term equation 2 by replacing the parameter $\theta_\star$ by an exponential moving average (EMA). In practical

terms, this simply means that, instead of defining $f_j = \mathcal{F}_{\theta_\star}(x_j)$, with $\theta_\star = \theta_k$ in equation 3, one sets $f_j = \mathcal{F}_{\theta_{\text{avg},k}}(x_j)$ where the EMA process $\{\theta_{\text{avg},k}\}_{k \geq 0}$ is defined through the recursion $\theta_{\text{avg},k} = (1 - \alpha\,\eta)\,\theta_{\text{avg},k-1} + \alpha\,\eta\,\theta_k$ where the coefficient $\alpha > 0$ controls the time-scale of the averaging process. The use of the EMA process $\{\theta_{\text{avg},k}\}_{k \geq 0}$ helps smoothing out the stochasticity of the process $\theta_k$. Similarly to Proposition 4.1, as $\eta \to 0$, the joint process $(\overline{\theta}_t^\eta, \overline{\theta}_{\text{avg},t}^\eta) \equiv (\theta_{[t/\eta]}^\eta, \theta_{\text{avg},[t/\eta]}^\eta)$ converges as $\eta \to 0$ to the solution of the following ordinary differential equation

$$\begin{cases} \dot{\overline{\theta}}_t = -\nabla\Big(\mathcal{L}(\overline{\theta}_t) + \lambda\,\mathcal{R}(\overline{\theta}_t, \overline{\theta}_{\text{avg},t})\Big) \\ \dot{\overline{\theta}}_{\text{avg},t} = -\alpha\,(\overline{\theta}_{\text{avg},t} - \overline{\theta}_t) \end{cases} \tag{5}$$

where the notation $\mathcal{R}(\overline{\theta}_t, \overline{\theta}_{\text{avg},t})$ designates the same quantity as the one described in equation 2, but with an emphasis on the dependency on the EMA process. At convergence $(\overline{\theta}_t, \overline{\theta}_{\text{avg},t}) \to (\overline{\theta}_\infty, \overline{\theta}_{\text{avg},\infty})$, one must necessarily have that $\overline{\theta}_\infty = \overline{\theta}_{\text{avg},\infty}$, confirming that, in the regime of small learning rate $\eta \to 0$, the *Mean Teacher* method converges, albeit often more rapidly, towards the same solution as the more standard $\Pi$-model. This indicates that the improved performances of the *Mean Teacher* approach sometimes reported in the literature are either not statistically meaningful, or due to poorly executed comparisons, or due to mechanisms not captured by the $\eta \to 0$ asymptotic. Indeed, several recently proposed consistency based SSL algorithms (BCG$^+$19; SBL$^+$20; XDH$^+$19) achieve state-of-the-art performance across diverse datasets without employing any exponential averaging processes. These results are achieved by leveraging more sophisticated data augmentation schemes such as *Rand-Augment* (CZSL19), *Back Translation* (ALAC17) or *Mixup* (ZCDLP17).

### 4.2 MINIMIZERS ARE HARMONIC FUNCTIONS

To understand better the properties of the solutions, we consider a simplified setting further exploited in Section 4.3. Assume that $\mathcal{F} : \mathcal{X} \equiv \mathbb{R}^D \to \mathbb{R}$ and $\mathcal{Y} \equiv \mathbb{R}$ and that, for every $y_i \in \mathcal{Y} \equiv \mathbb{R}$, the loss function $f \mapsto \ell(f, y_i)$ is uniquely minimized at $f = y_i$. We further assume that the data-manifold $\mathcal{M} \subset \mathbb{R}^D$ can be globally parametrized by a smooth and bijective mapping $\Phi : \mathbb{R}^d \to \mathcal{M} \subset \mathbb{R}^D$. Similarly to the Section 2, we consider a data-augmentation scheme that can be described as $\mathcal{S}_{\varepsilon\omega}(x) = \Phi(z + \varepsilon\omega)$ for $z = \Phi^{-1}(x)$ and a sample $\omega$ from a $\mathbb{R}^d$-valued centred and isotropic Gaussian distribution. We consider a finite set of labelled samples $\{x_i, y_i\}_{i \in \mathbf{I}_L}$, with $x_i = \Phi(z_i)$ and $z_i \in \mathbb{R}^d$ for $i \in \mathbf{I}_L$. We choose to model the large number of unlabelled data samples as a continuum distributed on the data manifold $\mathcal{M}$ as the push-forward measure $\Phi_\sharp \mu(dz)$ of a probability distribution $\mu(dz)$ whose support is $\mathbb{R}^d$ through the mapping $\Phi$. This means that an empirical average of the type $(1/|\mathcal{D}_U|) \sum_{i \in \mathbf{I}_u} \varphi(x_i)$ can be replaced by $\int \varphi[\Phi(z)]\,\mu(dz)$. We investigate the regime $\varepsilon \to 0$ and, similarly to Section 2, the minimization of the consistency-regularized objective

$$\mathcal{L}_L(\theta) + \frac{\lambda}{\varepsilon^2} \int_{\mathbb{R}^d} \mathbb{E}_\omega\Big\{\big\|\mathcal{F}_\theta[\mathcal{S}_{\varepsilon\omega}(\Phi(z))] - \mathcal{F}_\theta(\Phi(z))\big\|^2\Big\}\,\mu(dz). \tag{6}$$

For notational convenience, set $f_\theta \equiv \mathcal{F}_\theta \circ \Phi$. Since $\mathcal{S}_{\varepsilon\omega}[\Phi(z)] = \Phi(z + \varepsilon\,\omega)$, as $\varepsilon \to 0$ the quantity $\frac{1}{\varepsilon^2}\mathbb{E}_\omega\Big\{\big\|\mathcal{F}_\theta[\mathcal{S}_{\varepsilon\omega}(\Phi(z))] - \mathcal{F}_\theta(\Phi(z))\big\|^2\Big\}$ converges to $\|\nabla_z f_\theta\|^2$ and the objective function equation 6 approaches the quantity

$$\mathrm{G}(f_\theta) \equiv \frac{1}{|\mathcal{D}_L|} \sum_{i \in \mathbf{I}_L} \ell(f_\theta(z_i), y_i) + \lambda \int_{\mathbb{R}^d} \|\nabla_z f_\theta(z)\|^2\,\mu(dz). \tag{7}$$

A minimizer $f : \mathbb{R}^d \to \mathbb{R}$ of the functional G that is consistent with the labelled data, i.e. $f(z_i) = y_i$ for $i \in \mathbf{I}_L$, is a minimizer of the energy functional $f \mapsto \int_{\mathbb{R}^d} \|\nabla_z f_\theta(z)\|^2\,\mu(dz)$ subject to the constraints $f(z_i) = y_i$. It is the variational formulation of the Poisson equation

$$\begin{cases} \Delta f(z) = 0 & \text{for} \quad z \in \mathbb{R}^d \setminus \{z_i\}_{i \in \mathbf{I}_L} \\ f(z_i) = y_i & \text{for} \quad i \in \mathbf{I}_L. \end{cases} \tag{8}$$

Note that the solution does *not* depend on the regularization parameter $\lambda$ in the regime of $\varepsilon \to 0$: this indicates, as will be discussed in Section 4.3 in detail, that the generalization properties of consistency-based SSL methods will typically be insensitive to this parameter, in the regime of small data-augmentation at least. Furthermore, equation 8 shows that consistency-based SSL methods

are indeed based on the same principles as more standard graph-based approaches such as *Label Propagation* (ZG02): solutions are gradient/Laplacian penalized interpolating functions. In Figure 2, we consider the case where $D = d = 2$ with trivial mapping $\Phi(x) = x$. We consider labelled data situated on the right (resp. left) boundary of the unit square and corresponding to the label $y = 0$ (resp. $y = 1$). For simplicity, we choose the loss function $\ell(f, y) = \frac{1}{2}(f - y)^2$ and parametrize $\mathcal{F}_\theta \equiv f_\theta$ with a neural network with a single hidden layer with $N = 100$ neurons. As expected, the $\Pi$-model converges to the solution to the Poisson equation 8 in the unit square with boundary condition $f(u, v) = 0$ for $u = 0$ and $f(u, v) = 1$ for $u = 1$.

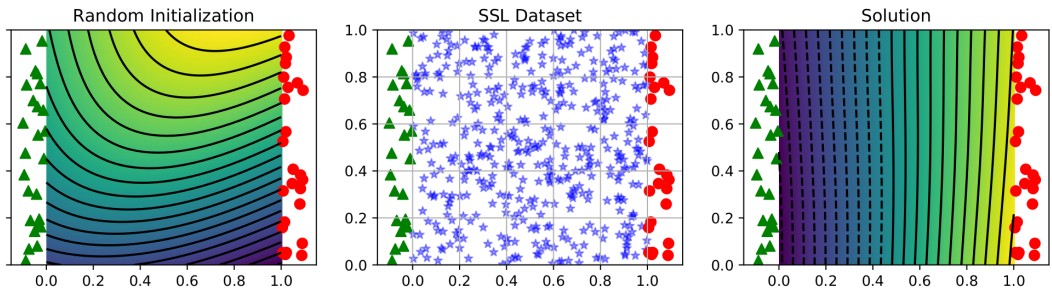

Figure 2: Labelled data samples with class $y = 0$ (green triangle) and $y = +1$ (red dot) are placed on the Left/Right boundary of the unit square. Unlabelled data samples (blue stars) are uniformly placed within the unit square. We consider a simple regression setting with loss function $\ell(f, y) = \frac{1}{2}(f - y)^2$. **Left:** Randomly initialized neural network. **Middle:** labelled/unlabelled data **Right:** Solution of $f$ obtained by training a standard $\Pi$-model. It is the harmonic function $f(u, v) = u$, as described by equation 8.

### 4.3    GENERATIVE MODEL FOR SEMI-SUPERVISED LEARNING

As has been made clear throughout this text, SSL methods crucially rely on the dependence structure of the data. The existence and exploitation of a much lower-dimensional manifold $\mathcal{M}$ supporting the data-samples is instrumental to this class of methods. Furthermore, the performance of consistency-based SSL approaches is intimately related to the data-augmentation schemes they are based upon. Consequently, in order to understand the mechanisms that are at play when consistency-based SSL methods are used to uncover the structures present in real datasets, it is important to build simplified and tractable generative models of data that **(1)** respect these low-dimensional structures and **(2)** allow the design of efficient data-augmentation schemes. Several articles have investigated the influence of the dependence structures that are present in the data on the learning algorithm (BM13; Mos16). Here, we follow the Hidden Manifold Model (HMM) framework proposed in (GMKZ19; GLK+20) where the authors describe a model of synthetic data concentrating near low-dimensional structures and analyze the learning curve associated to a class of two-layered neural networks.

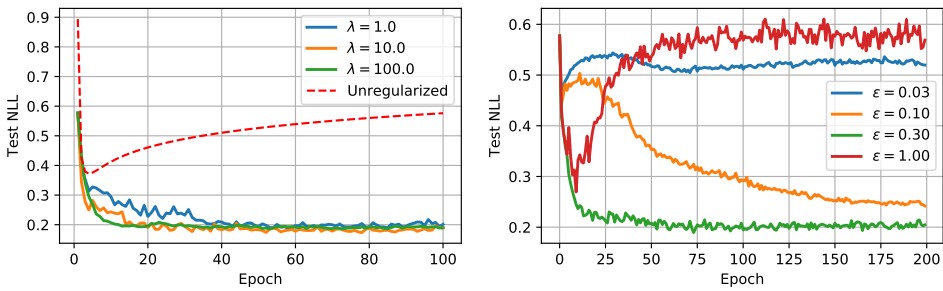

Figure 3: **Left:** For a fixed data-augmentation scheme, generalization properties for $\lambda$ spanning two orders of magnitude. **Right:** Influence of the quantity of the data-augmentation of the generalization properties.

**Low-dimensional structure:** Similarly to Section 4.2, assume that the $D$-dimensional data-samples $x_i \in \mathcal{X}$ can be expressed as $x_i = \Phi(z_i) \in \mathbb{R}^D$ for a fixed smooth mapping $\Phi : \mathbb{R}^d \to \mathbb{R}^D$. In other words, the data-manifold $\mathcal{M}$ is $d$-dimensional and the mapping $\Phi$ can be used to parametrize it. The mapping $\Phi$ is chosen to be a neural network with a single hidden layer with $H$ neurons, although other choices are indeed possible. For $z = (z^1, \ldots, z^d) \in \mathbb{R}^d$, set $\Phi(z) = A^{1 \to 2} \varphi(A^{0 \to 1} z + b^1)$ for matrices $A^{0 \to 1} \in \mathbb{R}^{H,d}$ and $A^{1 \to 2} \in \mathbb{R}^{D,H}$, bias vector $b^1 \in \mathbb{R}^H$ and non-linearity $\varphi : \mathbb{R} \to \mathbb{R}$ applied element-wise. In all our experiments, we use the ELU non-linearity. We adopt the standard normalization $A^{0 \to 1}_{i,j} = w^{(1)}_{i,j} / \sqrt{d}$ and $A^{1 \to 2}_{i,j} = w^{(2)}_{i,j} / \sqrt{H}$ for weights $w^{(k)}_{i,j}$ drawn i.i.d from a centred Gaussian distribution with unit variance; this ensures that, if the coordinate of the input vector $z \in \mathbb{R}^d$ are all of order $\mathcal{O}(1)$, so are the coordinates of $x = \Phi(z)$.

**Data-augmentation:** consider a data sample $x_i \in \mathcal{M}$ on the data-manifold. It can also be expressed as $x_i = \Phi(z_i)$. We consider the natural data-augmentation process which consists in setting $\mathcal{S}_{\varepsilon \omega}(x_i) = \Phi(z_i + \varepsilon \omega)$ for a sample $\omega \in \mathbb{R}^d$ from an isotropic Gaussian distribution with unit covariance and $\varepsilon > 0$. Crucially, the data-augmentation scheme respect the low-dimensional structure of the data: the perturbed sample $\mathcal{S}_{\varepsilon \omega}(x_i)$ belongs to the data-manifold $\mathcal{M}$ for any perturbation vector $\varepsilon \omega$. Note that, for any value of $\varepsilon$, the data-augmentation preserves the low-dimensional manifold: perturbed samples $\mathcal{S}_{\varepsilon \omega}(x_i)$ *exactly* lie on the data-manifold. The larger $\varepsilon$, the more efficient the data-augmentation scheme; this property is important since it allows to study the influence of the amount of data-augmentation.

**Classification:** we consider a balanced binary classification problem with $|\mathcal{D}_L| \geq 2$ labelled training examples $\{x_i, y_i\}_{i \in \mathbf{I}_L}$ where $x_i = \Phi(z_i)$ and $y_i \in \mathcal{Y} \equiv \{-1, +1\}$. The sample $z_i \in \mathbb{R}^d$ corresponding to the positive (resp. negative) class are assumed to have been drawn i.i.d from a Gaussian distribution with identity covariance matrix and mean $\mu_+ \in \mathbb{R}^d$ (resp. mean $\mu_- \in \mathbb{R}^d$). The distance $\|\mu_+ - \mu_-\|$ quantifies the hardness of the classification task.

**Neural architecture and optimization:** Consider fitting a two-layered neural network $\mathcal{F}_\theta : \mathbb{R}^D \to \mathbb{R}$ by minimising the negative log-likelihood $\mathcal{L}_L(\theta) \equiv (1/|\mathcal{D}_L|) \sum_i \ell[\mathcal{F}_\theta(x_i), y_i]$ where $\ell(f, y) = \log(1 + \exp[-y\,f])$. We assume that there are $|\mathcal{D}_L| = 10$ labelled data pairs $\{x_i, y_i\}_{i \in \mathbf{I}_L}$, as well as $|\mathcal{D}_U| = 1000$ unlabelled data samples, that the ambient space has dimension $D = 100$ and the data manifold $\mathcal{M}$ has dimension $d = 10$. The function $\Phi$ uses $H = 30$ neurons in its hidden layer. In all our experiments, we use a standard Stochastic Gradient Descent (SGD) method with constant learning rate and momentum $\beta = 0.9$. For minimizing the consistency-based SSL objective $\mathcal{L}_L(\theta) + \lambda\,\mathcal{R}(\theta)$, with regularization $\mathcal{R}(\theta)$ given in equation 2, we use the standard strategy (TV17) consisting in first minimizing the un-regularized objective alone $\mathcal{L}_L$ for a few epochs in order for the function $\mathcal{F}_\theta$ to be learned in the neighbourhood of the few labelled data-samples before switching on the consistency-based regularization whose role is to propagate the information contained in the labelled samples along the data manifold $\mathcal{M}$.

**Insensitivity to $\lambda$:** Figure 3 (Left) shows that this method is relatively insensitive to the parameter $\lambda$, as long as it is within reasonable bounds. This phenomenon can be read from equation 8 that does not depend on $\lambda$. Much larger or smaller values (not shown in Figure 3) of $\lambda$ do lead, unsurprisingly, to convergence and stability issues.

**Amount of Data-Augmentation:** As is reported in many tasks (CZM+18; ZCG+19; KYF20), tuning the amount data-augmentation in deep-learning applications is often a delicate exercise that can greatly influence the resulting performances. Figure 3 (Right) reports the generalization properties of the method for different amount of data-augmentation. Too low an amount of data-augmentation (i.e. $\varepsilon = 0.03$) and the final performance is equivalent to the un-regularized method. Too large an amount of data-augmentation (i.e. $\varepsilon = 1.0$) also leads to poor generalization properties. This is because the choice of $\varepsilon = 1.0$ corresponds to augmented samples that are very different from the distribution of the training dataset (i.e. distributional shift), although these samples are still supported by the data-manifold.

**Quality of the Data-Augmentation:** to study the influence of the *quality* of the data-augmentation scheme, we consider a perturbation process implemented as $\mathcal{S}_{\varepsilon \omega[k]}(x_i) = \Phi(z_i + \omega[k])$ for $x_i = \Phi(z_i)$ where the noise term $\omega[k]$ is defined as follows. For a *data-augmentation dimension* parameter $1 \leq k \leq d$ we have $\omega[k] = (\xi_1, \ldots, \xi_k, 0, \ldots, 0)$ for i.i.d standard Gaussian samples $\xi_1, \ldots, \xi_k \in \mathbb{R}$. This data-augmentation scheme only explores the first $k$ dimensions of the $d$-dimensional data-manifold: the lower $k$, the poorer the exploration of the data-manifold. As demonstrated on Figure

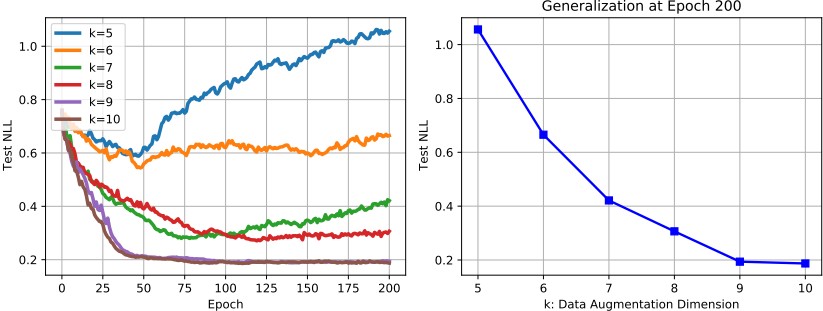

Figure 4: Learning curve test (NLL) of the $\Pi$-model with $\lambda = 10$ for different "quality" of data-augmentation. The data manifold is of dimension $d = 10$ in an ambient space of dimension $D = 100$. For $x_i = \Phi(z_i)$ and $1 \leq k \leq d$, the data-augmentation scheme is implemented as $\mathcal{S}_{\varepsilon\omega[k]}(x_i) = \Phi(z_i + \varepsilon\,\omega[k])$ where $\omega[k]$ is a sample from a Gaussian distribution whose last $(d - k)$ coordinates are zero. In other words, the data-augmentation scheme only explores $k$ dimensions out of the $d$ dimensions of the data-manifold. We use $\varepsilon = 0.3$ in all the experiments. **Left:** Learning curves (Test NLL) for data-augmentation dimension $k \in [5, 10]$ **Right:** Test NLL at epoch $N = 200$ (see left plot) for data-augmentation dimension $k \in [5, 10]$.

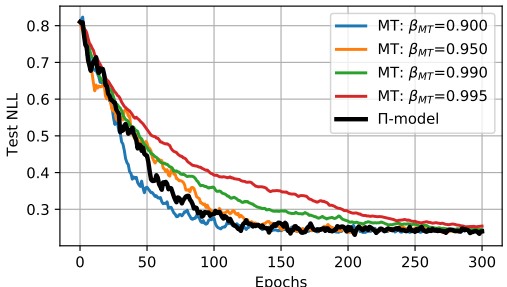

Figure 5: Mean-Teacher (MT) learning curves (Test NLL) for different values of the exponential smoothing parameter $\beta_{\mathrm{MT}} \in (0, 1)$. For $\beta_{\mathrm{MT}} \in \{0.9, 0.95, 0.99, 0.995\}$, the final test NLL obtained through the MT approach is identical to the test NLL obtained through the $\Pi$-model. In all the experiments, we used $\lambda = 10$ and used SGD with momentum $\beta = 0.9$.

4, lower quality data-augmentation schemes (i.e. lower values of $k \in [0, d]$) hurt the generalization performance of the $\Pi$-model.

**Mean-Teacher versus $\Pi$-model:** we implemented the Mean-Teacher (MT) approach with an exponential moving average (EMA) process $\theta_{\mathrm{avg},k} = \beta_{\mathrm{MT}}\,\theta_{\mathrm{avg},k-1} + (1 - \beta_{\mathrm{MT}})\,\theta_k$ for the MT parameter $\theta_{\mathrm{avg},k}$ with different scales $\beta_{\mathrm{MT}} \in \{0.9, 0.95, 0.99, 0.995\}$, as well as a $\Pi$-model approach, with $\lambda = 10$ and $\varepsilon = 0.3$. Figure 5 shows, in accordance with Section 4.1, that the different EMA schemes lead to generalization performances similar to a standard $\Pi$-model.

## 5 CONCLUSION

Consistency-based SSL methods rely on a subtle trade-off between the exploitation of the labelled samples and the discovery of the low-dimensional data-manifold. The results presented in this article highlight the connections with more standard methods such as Jacobian penalization and graph-based approaches and emphasize the crucial role of the data-augmentation scheme. The analysis of consistency-based SSL methods is still in its infancy and our numerical simulations suggest that the variant of the Hidden Manifold Model described in this text is a natural framework to make progress in this direction.

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
