# OpenReview forum: "On Data-Augmentation and Consistency-Based Semi-Supervised Learning"
_ICLR.cc/2021/Conference — ICLR 2021 Poster_

### Official Review · AnonReviewer3 · 2020-10-27
**ON DATA-AUGMENTATION AND CONSISTENCYBASED SEMI-SUPERVISED LEARNING**

**Rating:** 6
**Confidence:** 2

**Review:**

Summary: The authors analyze consistency-based models in specific settings where analytically tractable results can be obtained. They establish that leveraging more sophisticated data augmentation schemes is crucial to obtain huge gains when using consistency based models. Finally, they propose an extension of Hidden Manifold Model that incorporates data augmentation for understanding and experimenting with SSL methods.

Pros:
The paper is theoretically well grounded and the authors do a good job of connecting to previous works.

The paper is interesting and well written. It tries to explain the why consistency based method achieve good performance compared to other semi-supervised models. I find this relevant and helpful in understanding what makes these models work the way they do.

Cons:
The experiments are a bit weak only using simple synthetic settings.
Some of the claims in the paper are not backed up by experimental results.

Comments:
More experiments should be provided to establish some of the claims in the paper. Additionally, it will be good to see an experiment on a real dataset albeit small rather than simple synthetic experiments.

One of the claims of the paper is that mean teacher and simple Π-model approach share the same solutions in the regime where the data augmentations are small but advanced data augmentations performs better. It will be nice to demonstrate this with an experiment for the reader to better understand why this happens.

---

> ### Author Response · Authors · 2020-11-24
> **ON DATA-AUGMENTATION AND CONSISTENCYBASED SEMI-SUPERVISED LEARNING: answer to AnonReviewer3**
>
> Thank you for your comments:
> - "The experiments are a bit weak only using simple synthetic settings. Some of the claims in the paper are not backed up by experimental results." Thank you. We have now expanded further our investigations of the Hidden Manifold generative model to (1) show that the MT method and Π-model approach share the same solutions (2) the Hidden Manifold is a convenient framework for studying/controlling the quality+amount of the data-augmentation schemes. We have chosen to expand the synthetic example. This choice is motivated by our belief that the Hidden Manifold generative model is a good framework for (empirically) analysing SSL methods since it allows to quantify/control several key parameters (eg. quality/quantity of the Data-Augmentation) that are otherwise difficult to investigate on real datasets. We do believe that obtaining a theoretical understanding of SSL methods within the Hidden Manifold generative model is possible [eg. by considering the appropriate limiting regime when the dimensionality of the hidden layers goes to infinity], although it is (completely) out of the scope of the current manuscript.
> - "It will be nice to demonstrate this with an experiment for the reader to better understand why this happens." Thank you. We have followed your suggestion and added this empirical study and confirmed that MT/Pi-model do lead to similar solutions (at least in the regime considered in our manuscript).

---

### Official Review · AnonReviewer4 · 2020-10-28
**Insufficient experimental confirmation and there are some apparent contradictions in the proposed explanation, notably with MixUp**

**Rating:** 6
**Confidence:** 3

**Review:**

# Summary

This paper proposes a theoretical framework for understanding consistency-based semi-supervised learning. While establishing this framework based on the Hidden Manifold Model, this paper frames the SSL in the context of Manifold Tangent Classifiers.

# Pros

1. Formal understanding of SSL is indeed currently limited and theoretical works are needed.
2. The formalization of minimizers as harmonic function leads to the non-obvious prediction that SSL methods are insensitive, or at least very robust, to the weighting of the consistency loss $\lambda$. To me, this is the most important result of the paper.
3. The bibliography is well documented.

# Cons
1. The experimental verification of insensitivity to the weighting of consistency loss is unconvincing: it is done on a trivial low-dimensional toy dataset. It would be more convincing to take the GitHub code of one or more, possibly recent, SSL methods and vary $\lambda$ on real datasets.
2. I fail to see any other takeaway from the theoretical framework than the predicted insensitivity to $\lambda$.
3. Technically, if I understand correctly, the insensitivity to $\lambda$ is not a takeaway of the Hidden Manifold Model but of the Minimizers are Harmonic Functions.
4. I didn’t see where the claim “... demonstrate that the quality of the perturbations is key to obtaining reasonable SSL performances” has indeed been demonstrated.

# Questions and nits
1. “Several extensions such as the Mean Teacher (TV17) and the Virtual Adversarial Training (VAT) (MMIK18) schemes have been recently proposed and have been shown to lead to state-of-the-art results in many SSL tasks.” They are far from the current state of the art, especially so for few labels, this would have been true before the advent of MixMatch but the state of the art changed drastically with its introduction.
2. “... consider as well the tangent plane Tx to M at x.” Here I have a hard time visualizing it. Say the manifold M is a 2D Gaussian point cloud for example, what would be the tangent plane for a point x in that cloud?
3. Following on question 2, how is the tangent plane property exploited other than by defining an orthonormal basis on it? And why can’t an orthonormal basis of the Manifold space itself be enough if instead we assume it is dense?
4. “Enriching the set of data-augmentation degrees of freedom with transformations such as elastic deformation or non-linear pixel intensity shifts is crucial to obtaining a high-dimensional local exploration manifold.” This seems in direct contradiction with MixMatch results which does not use any sophisticated augmentation: just pixel shift, mirroring and random pixel-wise linear interpolation between samples and labels (MixUp).
5. Proposition 4.1: Here I was hoping for a prediction for your method. You mention the “sequence of processes converges weakly” and I was hoping it would explain why SSL techniques are much slower than fully supervised to converge. But then, it seems this statement is not exploited in any way other than saying that it does indeed converge to the solution of the ODE.
6. “This indicates that the improved performances of the Mean Teacher approach sometimes reported in the literature are either not statistically meaningful, or due to poorly executed comparisons, or due to mechanisms not captured by the η → 0 asymptotic.” This is vague, considering Mean Teacher outperforms VAT, would this mean that the last option holds (due to mechanisms not captured by the η → 0 asymptotic)? Just to clarify what the last option actually means: does it mean the proposed analysis relies on assumptions that don’t capture the reality of the phenomenon?
7. “These results are achieved by leveraging more sophisticated data augmentation schemes such as ... Mixup”. It seems odd to see MixUp being referred to as sophisticated (as in domain specific). MixUp is domain agnostic, it simply linearly combines a pair of samples and their labels. So in fact, it seems even simpler than a pixel shift since it’s linear.
8. “... with a neural network with a single hidden layer with N = 100 neurons”. What non-linearity was used? I didn’t seem to find it, or I may have missed it.
9. “Figure 3 (Left) shows that, contrary to several other types of regularizations such as weight-decay, this method is relatively insensitive to the parameter λ”. I didn’t see it. It indeed shows that (a) the method is relatively insensitive to the parameter λ in the context of the toy task but it doesn’t show that (b) the method is sensitive to other types of regularization such as weight-decay.

Note: Ultimately I must admit a lot of the maths are above my head and I have no idea whether they are correct or not, therefore I didn't comment on them and only focused on the parts that I understand. On the other hand, I'm pretty confident in my understanding of SSL techniques and MixUp.

=====POST-REBUTTAL COMMENTS========
I thank the authors for the response and the efforts in the updated draft. Most of my queries were clarified and I raised my rating accordingly. However, unfortunately, I still think a more realistic validation (e.g. on non-toy dataset) would benefit the paper.

---

> ### Author Response · Authors · 2020-11-24
> **On Data-Augmentation and Consistency-Based Semi-Supervised Learning: answer to AnonReviewer4**
>
> Thank you for these interesting comments that have allowed us, we hope, to better ground our manuscript within modern methodological SSL practices.
> - "The formalization of minimizers as harmonic function leads to the non-obvious prediction that SSL methods are insensitive, or at least very robust, to the weighting of the consistency loss \lambda. To me, this is the most important result of the paper." Thank you. To us, the main take-away of the manuscript is the use of the Hidden Manifold framework to spearhead the study of SSL methods (which is still extremely limited, contrarily to the fully supervised settings).
> - "The experimental verification of insensitivity to the weighting of consistency loss is unconvincing: it is done on a trivial low-dimensional toy dataset." : Thank you. We do not consider this example as trivial. Exploring a 10-dimensional manifold "hidden" in a D=100 dimensional ambient space is, we believe, challenging. The **intrinsic** dimension of the MNIST/CIFAR datasets, for example, are not that much higher. We could have ran a few experiments on more realistic datasets. We have chosen to continue with our synthetic example because of (1) tight compute constraints (2) our goal to illustrate the usefulness of the Hidden Manifold framework for studying SSL methods (3) the difficulty to control/quantify the data-augmentation schemes when applied to real datasets.
> - “I didn’t see where the claim ... demonstrate that the quality of the perturbations is key to obtaining reasonable SSL performances has indeed been demonstrated.”: thank you. We have now expanded the experimental part and numerically "quantified" that the quality [in our case, the dimensionality of the exploration of the latent manifold] directly influence the generalisation performance of the Pi-model. Although this can be characterized as "intuitive" (ie. better DA leads to better performace), having a modeling framework (i.e. Hidden Manifold generative model) that allows to investigate and empirically observe this phenomenon is, we believe, an important step for the study of SSL methods. As far as we are aware, no other article has followed this road for studying SSL methods.
> - "They are far from the current state of the art": thank you, this is an important point and we have clarified accordingly. Note nevertheless that the (sota) BYOL approach relies on mechanisms that are **very** similar to the MT/Pi-models [ie. basic consistency regularisation]. We believe that our framework could be useful to investigate SSL approaches such as the BYOL method.
> - "This seems in direct contradiction with MixMatch results which does not use any sophisticated augmentation: just pixel shift, mirroring and random pixel-wise linear interpolation between samples and labels (MixUp)": Thank You. There are many situations (eg. in other projects, we have ran many experiments in medical imaging settings) where elastic transformations are absolutely crucial [ie. MixMatch is far from enough for obtaining robust generalisation properties]. We do believe, though, that your point is indeed valid in many other situations. Conclusion: we need more theoretical investigations to understand when/why  MixMatch-type augmentations, for example, are enough in some situations, and far from enough in other settings.
> - "[...] I was hoping it would explain why SSL techniques are much slower [...]". thank you. Our results does *not* explain this phenomenon. It *is* a very interesting question. We do *not* have an explanation for it. Our framework is *unlikely* to easily lead to an understanding of this phenomenon.
> - "[...] This is vague, considering Mean Teacher outperforms VAT, would this mean that the last option holds [...]". Thank you. You are correct, our analysis relies on fix learning rate and in the limit when learning rate vanishes [ie. "fluid limit" asymptotic]. Standard practices such as adaptive learning rates / cyclic learning rates / momentums and other common optimisation practices are not covered by our analysis. We do agree that our result is only a (small) step towards understanding this class of methods.
> - "What non-linearity was used? I didn’t seem to find it, or I may have missed it." Thank you. We always used the ELU activation for keeping everything differentiable. It is now clarified in the text, thanks.
> - "I didn’t see it. It indeed shows that (a) the method is relatively insensitive to the parameter \lambda in the context of the toy task but it doesn’t show that (b) the method is sensitive to other types of regularization such as weight-decay." Thank you, it is a good point. We have removed this sentence (because of space considerations: we do believe that methods are often sensitive to the weight-decay parameter).
> - " maths are above my head [...] I only focused on the parts that I understand. " Thank you. your comments on the practical aspects of the manuscript *were* very useful to us.

---

### Official Review · AnonReviewer1 · 2020-10-29
**On Data-Augmentation and Consistency-Based Semi-Supervised Learning**

**Rating:** 6
**Confidence:** 4

**Review:**

This paper analyses the consistency-based SSL methods in settings  where  data lie a manifold of much lower dimension than the input space and obtains tractable results. The paper relates the analysis with Manifold Tangent Classifiers and shows that the quality of the perturbations plays a key role  to achieve a promising results in this set of SSL methods. Finally, the paper extends the Hidden Manifold Model by incorporating data-augmentation techniques and proposes a framework  to provide a direction for analyzing consistency-based SSL methods.

+This work might be useful for those who want to work in the theoretical part of SSL.

+The paper analytically discusses that the type of data augmentation plays a significant role in the performance  of the SSL models based on consistency regularization.

-While many points discussed in the paper are natural to me and well-discussed in the SSL literature (e.g.,  considering geometry of the data to develop an SSL algorithm or effect of the perturbation quality on the performance), relating and analyzing them with Manifold Tangent Classifiers (MTC) is interesting and new to me.  However, I still think that the theoretical part of the work is not strong enough and can be improved. The quality of the paper will be  improved if it uses MTC and provides some new results which do not exist in the SSL literature. This is because  MTC may not be the only approach to analyze these points.  For example, there are many other approaches based on optimal transport (e.g., Wasserstein Distances) that consider the geometry of the data and can be used to analyze the effect of perturbation on the SSL performance.  Then, someone may ask what is special in your approach to analysis consistency-based SSL methods in contrast to other tools/techniques?

-While this paper is mostly an analytical paper, providing more experiments on some claims discussed in the paper  (e.g., Mean Teacher method and Π-model approach share the same solutions in the cases where the data-augmentations are small)  is necessary and can improve the quality of the paper. Furthermore, authors may  use the exact same underlying model, training set-up,  and then try different types of data-augmentation methods on several recent and effective consistency-based SSL methods to show the differences on the performance experimentally. This can better contextualize the paper as we will know how much is the difference in terms of accuracy between different consistency-based SSL methods  with respect to perturbation, or other data-augmentation methods.

Generally,  I think this  paper provides a good direction for understanding the consistency based-SSL methods.

---

> ### Author Response · Authors · 2020-11-24
> **On Data-Augmentation and Consistency-Based Semi-Supervised Learning**
>
> Thank you for your comments.
> - "The quality of the paper will be improved if it uses MTC and provides some new results which do not exist in the SSL literature": to the best of our knowledge, a study of (1) the influence of the parameter \lambda that controls the trade-off between supervised and consistency loss terms (2) the equivalence of MT and Pi-model, and (3) [most importantly to our views] the use of Hidden Manifold generative framework for studying SSL, are new.
> - "This is because MTC may not be the only approach to analyze these points": agreed.
> - "someone may ask what is special in your approach to analysis consistency-based SSL methods in contrast to other tools/techniques?": Thank you. It allowed us to obtain a few results theoretical results, which is quite special to us. Furthermore, the Hidden Manifold framework, as we now empirically demonstrate at the very end of the article, allows to efficiently design controlled numerical experiments where the amount of DA can be accurately controlled + the quality of the exploration of the (latent) data-manifold can be quantified.
> - "there are many other approaches based on optimal transport (e.g., Wasserstein Distances) that consider the geometry of the data and can be used to analyze the effect of perturbation on the SSL performance." Thank you. We are not aware of these studies and have not managed to find appropriate references.
> - "providing more experiments on some claims discussed in the paper (e.g., Mean Teacher method and Π-model approach share the same solutions in the cases where the data-augmentations are small) is necessary and can improve the quality of the paper": thank you for this suggestion. We have now expanded further our investigations of the Hidden Manifold generative model to (1) show that the MT method and Π-model approach share the same solutions (2) the Hidden Manifold is a convenient framework for studying/controlling the quality+amount of the data-augmentation schemes.

---

### Decision · Program_Chairs · 2021-01-07
**Final Decision**

**Decision:**

Accept (Poster)

**Comment:**

This paper provides some theoretical perspective on the use of data augmentation in consistency regularization-based semi-supervised learning. The framework used in the paper argues that high-quality data augmentation should move along the data manifold. This generic view allows the paper's ideas to be applied across datasets (as opposed to image-specific data augmentation used in state-of-the-art semi-supervised learning algorithms). I am not aware of any other work raising these points, and indeed this paper is significant in that it provides a new and potentially useful perspective on the most performative semi-supervised learning approach. Reviewers agreed that the paper was clear and useful. The main concern was that the paper only included experiments in toy settings. Indeed, it would have been much more impactful to apply these ideas to state-of-the-art semi-supervised learning methods, but I think it can be excused given the theoretical focus of the work.